# Multi-population genetic algorithm with ER network for solving flexible job shop scheduling problems

Xiaoqiu Shi[1], Wei Long[2], Yanyan Li[2]*, Dingshan Deng[2]

**1** School of Manufacturing Science and Engineering, Southwest University of Science and Technology, Mianyang, China, **2** School of Mechanical Engineering, Sichuan University, Chengdu, China

* yyl_scu@163.com

**Data Availability Statement:** All relevant data are within the manuscript and its Supporting Information files.

**Funding:** This research was funded by the Science and Technology Department of Sichuan Province

## Abstract

A genetic algorithm (GA) cannot always avoid premature convergence, and multi-population is usually used to overcome this limitation by dividing the population into several sub-populations (sub-population number) with the same number of individuals (sub-population size). In previous research, the questions of how a network structure composed of sub-populations affects the propagation rate of advantageous genes among sub-populations and how it affects the performance of GA have always been ignored. Therefore, we first propose a multi-population GA with an ER network (MPGA-ER). Then, by using the flexible job shop scheduling problem (FJSP) as an example and considering the total individual number (TIN), we study how the sub-population number and size and the propagation rate of advantageous genes affect the performance of MPGA-ER, wherein the performance is evaluated by the average optimal value and success rate based on TIN. The simulation results indicate the following regarding the performance of MPGA-ER: (i) performance shows considerable improvement compared with that of traditional GA; (ii) for an increase in the sub-population number for a certain TIN, the performance first increases slowly, and then decreases rapidly; (iii) for an increase in the sub-population size for a certain TIN, the performance of MPGA-ER first increases rapidly and then tends to remain stable; and (iv) with an increase in the propagation rate of advantageous genes, the performance first increases rapidly and then decreases slowly. Finally, we use a parameter-optimized MPGA-ER to solve for more FJSP instances and demonstrate its effectiveness by comparing it with that of other algorithms proposed in other studies.

## 1. Introduction

The genetic algorithm (GA) is a widely used evolutionary algorithm [1–3]. When solving problems with GA, feasible solutions are first encoded in individuals, which can then be conveniently processed by operators (e.g., crossover and mutation). A certain number of individuals constitute a population, wherein the individuals communicate with each other, and advantageous genes propagate and accumulate in the population. Finally, a satisfactory solution is

(Grant No. 20CXRC0097) to YL. The funder had no role in study design, data collection and analysis, decision to publish, or preparation of the manuscript.

**Competing interests:** The authors have declared that no competing interests exist.

obtained. To avoid premature convergence, which is the main disadvantage of standard GA, a multi-population method that is effective in improving GA is used. This results in a different algorithm: the multi-population genetic algorithm (MPGA) [4, 5]. The MPGA divides the population of a standard GA into $N$ sub-populations (the sub-population number is denoted by $N$) that each include the same number of individuals (the sub-population size is denoted by $S$). Here, elite individuals migrate among different sub-populations with a certain frequency ($R$), thereby resulting in advantageous genes propagating among and within sub-populations. Concepts such as distributed GAs [10] and parallel GAs [11] are similar to MPGA because they maintain several sub-populations during the process of evolution. If sub-populations are regarded as nodes, and the migrations of elite individuals between them as edges connecting different nodes, then MPGA can be depicted as a network. Studies have shown that a network structure formed by the edges of a network has a significant impact on its behavior. For example, cooperative evolution [6, 7], risk propagation [8], and distress propagation [9] have shown that network structures indeed have a significant impact on their behaviors. Similarly, a network structure formed by the propagation mode of advantageous genes among sub-populations of MPGA also affects the behavior of the MPGA itself. In fact, some scholars have studied GA and other evolutionary algorithms by using various networks. For example, in [10], different crossovers were used in different sub-populations, and a GA with a better performance was proposed by using a regular network. In [11], a GA was proposed using regular networks, such as fully connected networks, and the influence of migration frequency and other parameters on its performance was studied. In [12], the selection pressures of evolutionary algorithms were studied using one- and two-dimensional regular networks. In [13], graph-based evolutionary algorithms were proposed using networks such as complete graphs, complete bipartite graphs, n-cycles, and trees. In the aforementioned studies, scholars often used regular networks to study GA and other evolutionary algorithms; however, some studies have shown that real networks (e.g., the Internet network) are mostly irregular scale-free networks [14]. Furthermore, some scholars have used scale-free networks to study evolutionary algorithms. For example, in [15], the authors studied the propagation dynamics behavior of an evolutionary algorithm using scale-free networks from a theoretical perspective (it was measured by takeover time, i.e., the duration it takes until advantageous genes fill the whole population), thereby revealing the influencing mechanisms of different network structural parameters on the selection pressures of the algorithm. However, only two fitness values (0 and 1) were considered in that study, which limits the practical applications of its results. In [16], the authors indicated that evolutionary algorithms designed by scale-free networks do not perform better than those designed by regular networks. In [18], the authors stated that the performance of evolutionary algorithms designed by scale-free networks is worse than those designed by random and small-world networks [19] while solving the double-objective problem. An evolutionary algorithm designed by a scale-free network tends to perform better when the number of optimization objectives increases. In [20], a multi-objective evolutionary algorithm with improved performance was designed using control graphs. Additionally, in our previous research [17], we used seven different networks to design MPGAs and then studied how different network topologies affected the performance of the MPGAs.

As summarized in Table 1, in the existing literature, mainly regular [10–13], scale-free [15–18], random [18], and small-world networks [18] have been used to study GA or other evolutionary algorithms. In addition, these networks have been mostly used to control propagation behaviors of advantageous genes between individuals [13, 15, 16, 18, 20], and have rarely been used to control the propagation behaviors of advantageous genes among sub-populations. Therefore, it is still not clear how the propagation behavior (this study mainly refers to the propagation rate) of advantageous genes among sub-populations affects the performance of

**Table 1. Examples of evolutionary algorithms using different networks.**

| Studies | Networks | Descriptions |
|---|---|---|
| Herrera et al. [10] | ▪ Regular hypercube. | ▪ Apply different crossover operators to different sub-populations to distinguish between these sub-populations.<br>▪ Use the network to control sub-populations. |
| Cantu-Paz [11] | ▪ Fully connected topology.<br>▪ Uni- and bi-directional rings.<br>▪ Regular hypercube. | ▪ Apply different topology to design GAs, and then study how some parameters, such as the number of populations, their size and the migration rate affect the performance of these GAs.<br>▪ Use the network to control sub-populations. |
| Giacobini et al. [12] | ▪ One-dimensional lattice.<br>▪ Two-dimensional lattice. | ▪ Apply regular one- and two-dimensional (2-D) lattices to design evolutionary algorithms, and then study their selection pressures.<br>▪ Use the network to control sub-populations. |
| Bryden et al. [13] | ▪ Twenty-six types of networks, such as complete graph, complete bipartite graph, $n$-cycle and tree. | ▪ Apply different graphs to limit possible crossover partners in a population, and then study the performance of differently obtained evolution algorithms.<br>▪ Use the network to control individuals. |
| Payne et al. [15] | ▪ Scale-free network | ▪ Apply scale-free networks with different parameters to limit possible crossover partners in a population, and then reveal the influencing mechanisms of different network structural parameters on the selection pressures of the evolution algorithm.<br>▪ Use the network to control individuals. |
| Giacobini et al. [16] | ▪ Scale-free network<br>▪ Small-world network | ▪ Apply scale-free and small-world networks to limit possible crossover partners in a population, revealing that evolutionary algorithms designed by scale-free networks are not better than those designed by regular networks.<br>▪ Use the network to control individuals. |
| Kirley et al. [18] | ▪ Scale-free network<br>▪ Small-world network<br>▪ Random network | ▪ Apply scale-free, small-world and random networks to limit possible crossover partners in a population, and then study the performance of differently obtained algorithms on multi-objective optimization problems.<br>▪ Use the network to control individuals. |
| Mateo et al. [20] | ▪ Directed domination graphs | ▪ Use directed domination graphs to represent the domination relations between individuals in the population, obtaining a multi-objective evolutionary algorithm with an improved performance.<br>▪ Use the network to control individuals. |
| Shi et al. [17] | ▪ Seven kinds of networks, including scale-free, block-diagonal, centralized, random, hierarchical, local and small-world networks. | ▪ Apply seven networks to design MPGAs, and then study how different network topologies affect the performance of MPGAs.<br>▪ Use the network to control sub-populations. |

MPGA. Meanwhile, existing literature [12, 15] mainly includes studies on the propagation behaviors of advantageous genes in evolutionary algorithms from a theoretical perspective. However, the relationship between the propagation rate of advantageous genes and the performance of algorithms while solving practical problems was rarely studied. Moreover, existing

literature rarely includes studies on how parameters $N$ and $S$ affect MPGA while considering a certain total individual number (TIN) [4]. Therefore, by using the example of the flexible job shop scheduling problem (FJSP) [21], this work studies how parameters $N$, $S$, and the propagation rate of advantageous genes among sub-populations affect the performance of MPGA. In recent years, some scholars have used MPGA to solve FJSP [4, 22]; however, the number of sub-populations in their studies was very limited.

To this end, first, the network generated by the ER model [23] and the migration frequency are used to control the propagation behaviors of advantageous genes among sub-populations. Then, for simplicity, easy realization, and a wide applicability of GA, we use the ER network to design the MPGA, obtaining a multi-population genetic algorithm with the ER network (MPGA-ER). In [17], we used seven networks, including the ER network, to design the MPGA, whereas the connection probability ($P$) of the ER model was a constant. We found that the propagation rate of advantageous genes among sub-populations was limited by these network topologies; hence, how the propagation rate affects the performance of MPGA over a wider range is still not clear. Fortunately, when the connection probability of the ER model changes from 0 to 1, the corresponding network gradually transforms from a graph that only includes isolated points to a complete graph. Therefore, the propagation rate of the advantageous genes changes from considerably low to high. $P$ and $R$ can conveniently be used to control the propagation rate of advantageous genes over a wide range. Accordingly, as an extension of our previous research [17], the ER model is used in this study to design the MPGA, thereby revealing how the propagation rate affects the performance of the MPGA over a wide range. Second, an evaluation index, the Hamming distance evaluator (HDE) [17], was used to evaluate the propagation rate of advantageous genes. Third, MPGA-ER is used to solve an FJSP instance, and how the propagation rate of advantageous genes affects the performance of MPGA-ER is also studied. Meanwhile, the influence of parameters $N$ and $S$ on the performance of MPGA-ER based on TIN (which is defined as the total individual number used by the algorithm during an entire searching process) is also studied. Finally, the parameter-optimized MPGA-ER is used to solve more FJSP instances. A comparison of this MPGA-ER with other algorithms proposed in other studies demonstrates its effectiveness.

## 2. Basic problems and solution methodologies

### 2.1 Flexible job shop scheduling problem

For the realization of a cost-effective and reliable production, many scheduling problems have been studied in recent years, such as the flexible job shop scheduling problem [4, 17], job shop scheduling problem [24], flow shop scheduling problem [35], and stable maintenance tasks scheduling [36]. Among them, FJSP was initially proposed by Brucker et al. [21], wherein an operation can be processed on multiple machines. Therefore, this FJSP becomes a more complex NP-hard problem than the basic job shop scheduling problem, in which an operation can only be processed on a single machine [24]. Owing to its complexity, FJSP has attracted the attention of several scholars, and many corresponding mathematical models have been established [4, 17, 25–27]. Hence, we investigated FJSP in this study. In comparison to these models [4, 17], FJSP can be described as follows:

There are $n$ jobs ($J_1$, $J_2$, . . ., $J_n$) that can be processed on $m$ machines ($M_1$, $M_2$, . . ., $M_m$), whereas the $i^{th}$ job ($J_i$) comprises $n_i$ operations ($O_{i1}$, $O_{i2}$, . . ., $O_{ini}$). The total number of operations of all jobs is denoted by $J_t$. The $j^{th}$ operation of the $i^{th}$ job ($O_{ij}$) can be processed on a candidate machine set ($S_{ij}$). The processing time of $O_{ij}$ on the $k^{th}$ machine ($M_k$) is denoted by $P_{ijk}$. The processing times of an operation on different machines can be different; all machines are available at the starting time, regardless of machine failure. All materials were prepared at the

starting time, regardless of the handling time. An operation at a given time can only be processed on one machine without interruption. One machine can only process one operation at a time. The processing order of the given job is known and fixed (this is known as a processing constraint). FJSP comprises two sub-problems: (i) machine selection, which refers to the selection of suitable machines for each operation and (ii) operation arrangement, which refers to the arrangement of a reasonable processing sequence that meets the processing constraints for all the operations assigned to a machine. Therefore, FJSP must determine the start and completion times of each operation on the selected machine under the premise of meeting the processing constraints to meet the given targets. Common goals of FJSP include minimizing the maximum completion time, minimizing the machine load, etc. For simplicity, the most widely used goal, minimizing the maximum completion time, is adopted here. The mathematical model of FJSP is as follows [4, 17]:

$$\min F_{\max} = \min(\max_j(F_{ij})) \tag{1}$$

*s.t.*

$$F_{ij} - F_{i(j-1)} - P_{ijk} \times X_{ijk} \geq 0, \forall i, j, k \tag{2}$$

$$\sum_{k \in S_{ij}} X_{ijk} = 1 \wedge F_{ijk} - B_{ijk} = P_{ijk}, \forall i, j \tag{3}$$

$$F_{i'j'k} \leq B_{ijk} \vee F_{ijk} \leq B_{i'j'k}, \forall i', j' \neq i, j \tag{4}$$

$$i, j, k, i', j' \in \{1, 2, 3, \ldots\} \tag{5}$$

$$X_{ijk} \in \{0, 1\}, \forall i, j, k \tag{6}$$

$$F_{ij} \geq 0, \forall i, j \tag{7}$$

$$P_{ijk} \geq 0, \forall i, j, k \tag{8}$$

$$F_{ijk} \geq 0, \forall i, j, k \tag{9}$$

$$B_{ijk} \geq 0, \forall i, j, k \tag{10}$$

In Eq (1), which is an objective function, "min ()" and "max ()" represent the functions that set the minimum and maximum values, respectively. $F_{max}$ represents the maximum completion time, and $F_{ij}$ represents the completion time of $O_{ij}$. Eq (2) represents the processing constraints: $F_{i(j-1)}$ represents the completion time of the previous operation of $O_{ij}$, and symbol "$\forall$" represents "any." When $O_{ij}$ is processed on $M_k$, $X_{ijk}$ is set to 1; otherwise, it is set to 0. Eq (3) guarantees that an operation can only be processed on one machine at a time without interruption, where $F_{ijk}$ and $B_{ijk}$ represent the start and the completion times of $O_{ij}$ on $M_k$, respectively, and symbol "$\wedge$" represents a "Logical AND". Eq (4) indicates that one machine can only process one operation at a time, where symbol "$\vee$" represents a "Logical OR". Eqs (5)–(10) denote the domains of these variables. For clarity, the symbols used in the aforementioned FJSP model are summarized in Table 2.

**Table 2. Symbols of FJSP.**

| Symbols | Descriptions |
|---|---|
| $n$ | The number of jobs. |
| $J_i$ | The $i^{th}$ job, $i = 1, \ldots, n$. |
| $n_i$ | The number of the $i^{th}$ job. |
| $O_{ij}$ | The $j^{th}$ operation of the $i^{th}$ job, $j = 1, 2, \ldots n_i$. |
| $J_t$ | The total number of operations of all jobs. |
| $m$ | The number of machines. |
| $M_k$ | The $k^{th}$ machine, $k = 1, \ldots, m$. |
| $S_{ij}$ | The candidate machine set of $O_{ij}$, wherein $O_{ij}$ can be processed on these machines. |
| $P_{ijk}$ | The processing time of $O_{ij}$ on $M_k$. |
| $F_{max}$ | The maximum completion time of all jobs. |
| $F_{ij}$ | The completion time of $O_{ij}$. |
| $F_{i(j-1)}$ | The completion time of the previous operation of $O_{ij}$. |
| $F_{ijk}$ | The start time of $O_{ij}$ on $M_k$. |
| $B_{ijk}$ | The completion time of $O_{ij}$ on $M_k$. |
| min () | The function that sets the minimum value. |
| max () | The function that sets the maximum value. |
| $\forall$ | Denotes "any". |
| $X_{ijk}$ | When $O_{ij}$ is processed on $M_k$, it is set to 1, otherwise it is set to 0. |
| $\wedge$ | Denotes "Logical AND". |
| $\vee$ | Denotes "Logical OR". |

Table 3 presents an FJSP example with two jobs and five machines, wherein symbol "/" indicates that the corresponding operation cannot be processed on the corresponding machine.

## 2.2 Multi-population genetic algorithm with ER network

A network can simply be regarded as a graph $G = (V, E)$, where $V$ represents the node set and $E$ is the edge set. If there exists an edge ($e_{ij}$) between nodes $V_i$ and $V_j$, then $e_{ij} \in E$. The ER model is a classic model for generating random networks. The generated ER network is used in this study to design the MPGA, thereby obtaining the MPGA-ER. The ER model can be described as follows: (i) given a graph $G_0$ with $N$ isolated nodes; (ii) given a predefined connection probability $P$; (iii) for each pair of nodes ($V_i$ and $V_j$) in $G_0$, if rand(1) < $P$, the two nodes are connected by using an edge $e_{ij}$ (multiple edges and rings are not allowed), where "rand(1)" refers to a random number between 0 and 1; and (iv) if all pairs of nodes adhere to "(iii)", an ER network with a given $P$ is obtained. As mentioned above, nodes represent the sub-populations of MPGA-ER, and edges are the migrations of elite individuals among sub-populations. In Fig 1, an MPGA-ER with five sub-populations, each of which includes four individuals is

**Table 3. Example of FJSP.**

| Jobs | Operations | $M_1$ | $M_2$ | $M_3$ | $M_4$ | $M_5$ |
|---|---|---|---|---|---|---|
| $J_1$ | $O_{11}$ | 2 | 5 | / | 5 | / |
| | $O_{12}$ | 1 | 5 | 9 | 7 | 4 |
| $J_2$ | $O_{21}$ | / | 1 | / | 4 | / |
| | $O_{22}$ | / | / | 1 | 5 | 5 |
| | $O_{23}$ | 5 | / | / | 2 | / |

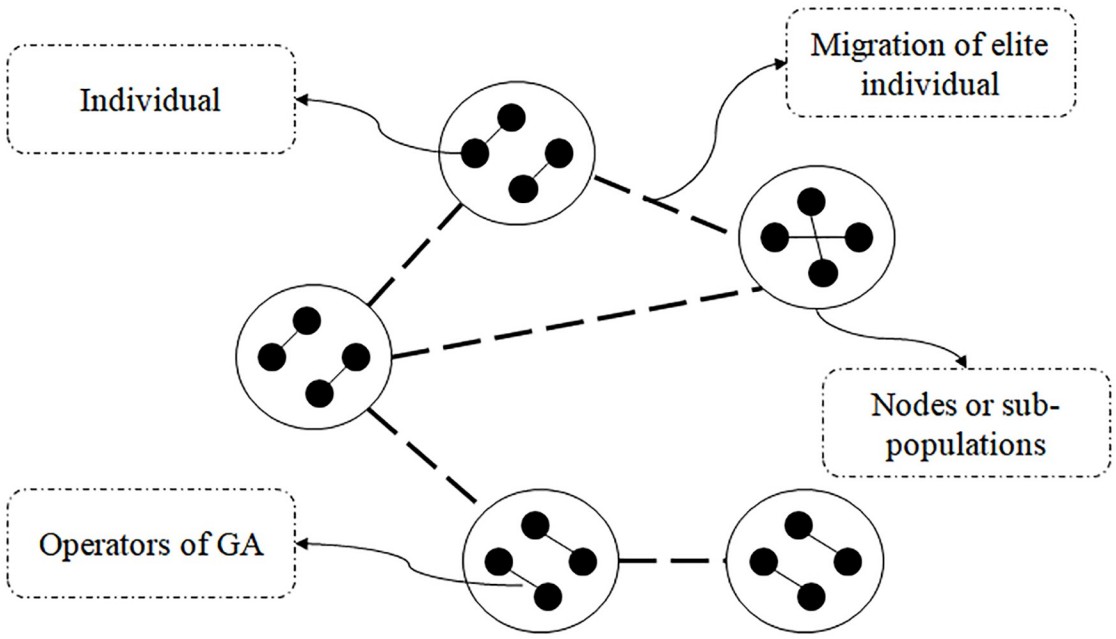

**Fig 1. Schematic diagram of MPGA-ER.**

illustrated, wherein advantageous genes are propagated among these sub-populations through the migrations of elite individuals.

The details of the MPGA-ER are as follows.

**Step 1**: This is the initialization step. Here, each of the $N$ sub-populations, each comprising $S$ individuals, is randomly initialized according to the coding rules, which are described later.

**Step 2**: If the algorithm has executed the given maximum number of iterations ($I_{max}$), exit the execution, and output the best solution for all sub-populations; otherwise, go to **Step 3**.

**Step 3**: For each sub-population, the fitness of each individual is decoded by the decoding algorithm, which is described later, and elite individuals are selected and stored in the elite set.

**Step 4**: For each sub-population, a standard competition selection of GA is used to generate the next sub-population, thereby keeping the sub-population size unchanged.

**Step 5**: For each sub-population, individuals are randomly divided into pairs that will undergo a crossover, which is described below, thereby obtaining the next generation of each sub-population.

**Step 6**: For each sub-population, a mutation, which is described below, occurs according to the mutation probability ($P_r$).

**Step 7**: Repeat **Steps** 3 to 6 $N$ times and obtain an elite set with $N$ elite individuals.

**Step 8**: If rand(1) $> = P_c$, go to **Step 9**, otherwise go to **Step 2**. Here, "rand(1)" refers to a random number between 0 and 1, and $P_c$ can be expressed as:

$$P_c = \left(\frac{I_{max} - I_{now}}{I_{max}}\right)^R, \tag{11}$$

where $I_{now}$ represents the current number of iterations of the algorithm, and $R$ is a term used to control the migration frequency of elite individuals among sub-populations. For $R = 0$, there are no migrations of elite individuals among sub-populations, and the greater the value of $R$, the higher the migration frequency of elite individuals.

**Step 9**: Randomly select a sub-population ($V_i$), find all neighboring sub-populations of $V_i$ according to the given ER network, find the best elite individual of all selected neighboring sub-populations (including $V_i$ itself), place the best elite individual randomly into all selected neighboring sub-populations, and return to **Step 2**.

## 2.3 MPGA-ER for solving FJSP

The main operators for solving FJSP using an MPGA-ER are coding, decoding, crossover, and mutation. They are explained as follows:

**2.3.1 Coding.** Integer coding described in [27] is used to generate an individual. It comprises of two parts: machine coding and operation coding. (i) Machine coding: it is represented as an integer string of length $J_t$, for which the position of integer represents the operation, and the integer itself determines which machine of the candidate machine set is selected. For example, a machine coding for the FJSP example in Table 3 is [3 1 2 3 2]. This FJSP has five operations; therefore, the machine coding string has exactly five integers. The first integer, 3, means that $O_{11}$ selects the third machine from its candidate machine set, $S_{11} = \{M_1, M_2, M_4\}$, that is, $M_4$ rather than $M_3$, and so on. (ii) Operation coding: it is also expressed as an integer string of length $J_t$, for which the position of the integer represents the processing order, and the integer itself represents the job number. The operation coding of the FJSP example shown in Table 3 is [2 1 2 1 2]. $J_2$ comprises of 3 operations; therefore, integer 2 appears exactly three times. The first integer, 2, denotes that $O_{21}$ will be processed first, and so on. The machine and operation coding are then combined to represent individuals such as [3 1 2 3 2 2 1 2 1 2].

**2.3.2 Decoding.** The description of the decoding algorithm [4] adopted in this study is as follows.

*Step 1*: Initialize a matrix (**F**) containing six columns and $J_t$ rows. Each row represents an operation. According to the coding rules, the machine selected for each operation and the corresponding processing time can easily be obtained. For example, consider the aforementioned individual, [3 1 2 3 2 2 1 2 1 2], as an example. The corresponding **F** is [2 1 4 4 0 0; 1 1 4 5 0 0; 2 2 5 5 0 0; 1 2 1 1 0 0; 2 3 4 2 0 0]. The first row, [2 1 4 4 0 0], means that $O_{21}$ is processed on $M_4$, and $P_{214}$ is 4, and the start and completion times of the operation are unknown (represented by 0s).

*Step 2*: Consider each row of **F**. If the operation represented by the row (set to $O_{ij}$) is the first operation of $J_i$ and no other operations are arranged on the machine selected by $O_{ij}$ (set it to $M_k$), then $B_{ijk} = 0$, and $F_{ijk} = B_{ijk} + P_{ijk}$. Store $B_{ijk}$ and $P_{ijk}$ in columns 5 and 6 of **F**, respectively. If $O_{ij}$ is the first operation of $J_i$ and other operations have been arranged on $M_k$, then all free intervals on $M_k$ denoted by $[s_q, e_q]$ ($q = 1, 2, \ldots$) can be found. Find the first interval with a length greater than or equal to $P_{ijk}$, then $B_{ijk} = s_q$, and $F_{ijk} = s_q + P_{ijk}$. If $O_{ij}$ is not the first operation of $J_i$, and no other operation is arranged on $M_k$, then $B_{ijk} = F_{ij-1k}$, and $F_{ijk} = B_{ijk} + P_{ijk}$. If $O_{ij}$ is not the first operation of $J_i$, and other operations have been arranged on $M_k$, then find all free intervals of $M_k$. Consider each free interval, $[s_q, e_q]$, in turn, if $e_q\text{-}s_q >= P_{ijk}$ and $F_{ij-1k} <= s_q$, then $B_{ijk} = s_q$, $F_{ijk} = B_{ijk} + P_{ijk}$; if $e_q\text{-}s_q >= P_{ijk}$, $F_{ij-1k} >= s_q$ and $e_q\text{-} F_{ij-1k} >= P_{ijk}$, then $B_{ijk} = F_{ij-1k}$, $F_{ijk} = B_{ijk} + P_{ijk}$ (The last interval is infinite; therefore, we can always find an interval that satisfies one of these conditions). For example, if the individual,

[3 1 2 3 2 2 1 2 1 2], is decoded by the above algorithm, then **F** = [2 1 4 4 0 4; 1 1 4 5 4 9; 2 2 5 5 4 9; 1 2 1 1 9 10; 2 3 4 2 9 11], representing a schedule. The Gantt chart of the schedule represented by **F** in the example in Table 3 is shown in Fig 2.

**2.3.3 Crossover.** According to the characteristics of the aforementioned integer coding, crossover can be divided into two stages: machine and operation crossover. (i) Machine crossover: two machine codings (Parent M-1 and Parent M-2) are randomly selected; two integers less than $J_t$ are randomly generated, and they are considered to be the points of the two-point crossover (as shown in Fig 3). In Fig 3, Parent M-1 [2 2 1 2 5 2 2 4] and Parent M-2 [1 1 2 2 3 4 5 5] are crossed. If the two randomly selected integers are less than $J_t$ and are 3 and 6, respectively, the integers of Parent M-1 between the 3$^{rd}$ position and 6$^{th}$ position, that is, (1 2 5 2), and the integers of Parent M-2 between the 3$^{rd}$ position and 6$^{th}$ position, that is, (2 2 3 4), are exchanged. Thus, Offspring M-1 and Offspring M-2 obtained using the two-point crossover are [2 2 2 2 3 4 2 4] and [1 1 1 2 5 2 5 5], respectively. (ii) Operation crossover [27]: two operation codings (Parent O-1 and Parent O-2) are randomly selected the jobs are randomly divided into two groups (Group 1 and Group 2); the integers of Parent O-1 (Parent O-2) which belong to Group 1 are copied to Offspring O-1 (Offspring O-2), where their positions are preserved; the integers of Parent O-2 (Parent O-1) which belong to Group 2 are copied to Offspring O-1 (Offspring O-2), where their order is preserved (as shown in Fig 4). In Fig 4, Parent O-1 [1 2 1 2 3 3 3 4] and Parent O-2 [2 1 3 2 1 4 3 3] are crossed. Jobs 2 and 4 are considered to be Group 1, and the remaining jobs Group 2. First, integers (2, 2, 4), which belong to Group 1 of Parent O-1 (Parent O-2), are copied to Offspring O-1 (Offspring O-2), where the positions are preserved. Then integers (1, 1, 3, 3, 3) of Parent O-1, which belong to Group 2 are copied to Offspring O-2, where the order is preserved. Finally, integers (1, 3, 1, 3, 3) of Parent O-2, which belong to Group 2, are copied to Offspring O-1, where the order is preserved.

**2.3.4 Mutation.** Mutation is also divided into two stages: machine and operation mutation. (i) Machine mutation: randomly select some individuals according to $P_r$; for each individual, an integer $r$ less than $J_t$ is randomly generated to represent a position; for this position, an integer smaller than the number of the corresponding candidate machines is randomly generated, and the generated integer is placed in the selected position. (ii) Operation mutation:

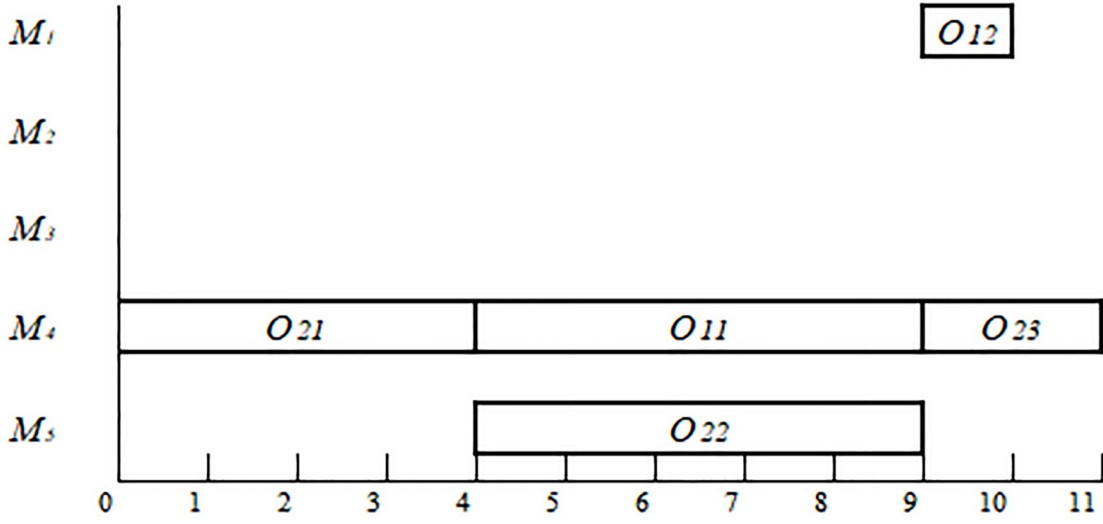

**Fig 2. The Gantt chart of a schedule for the example in Table 3.**

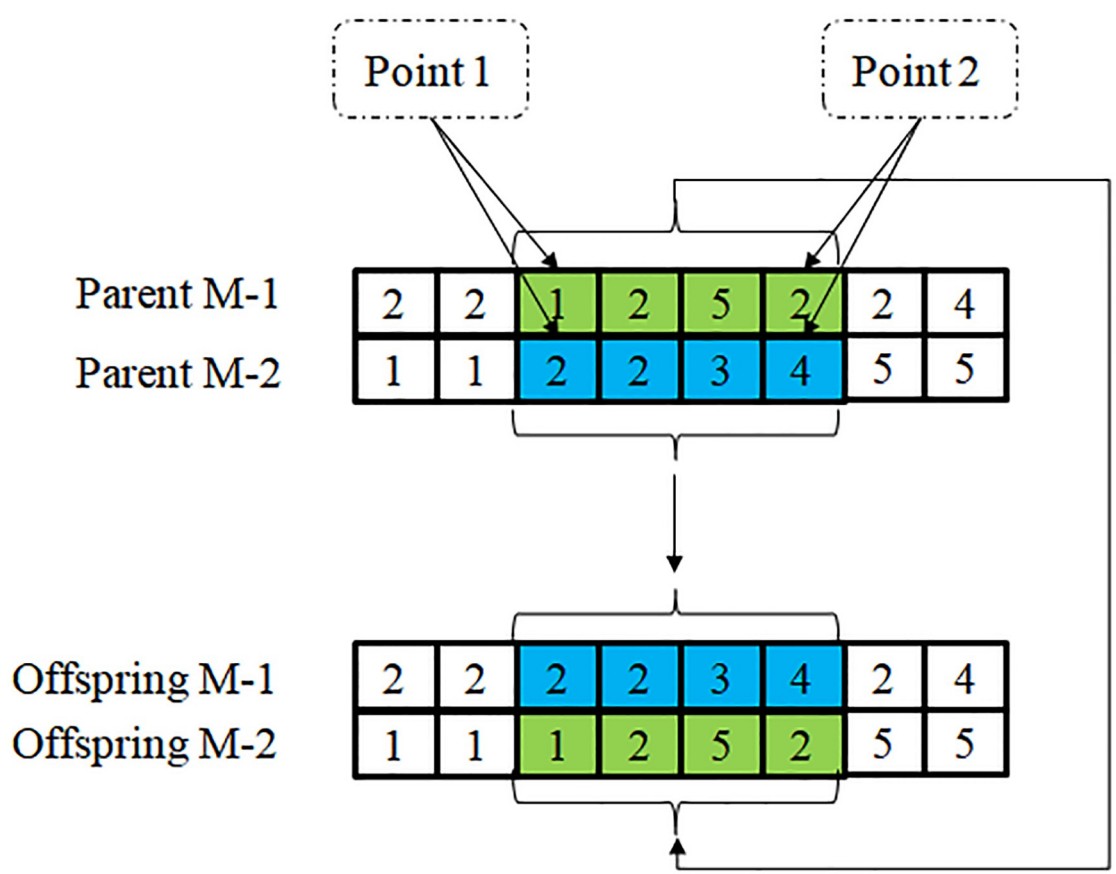

**Fig 3. Schematic diagram of two-point crossover.**

some individuals are randomly selected according to $P_r$; for each individual, an integer $r'$ smaller than $J_t/2$ is randomly generated; two integers smaller than $J_t$ representing two positions are generated randomly; the two integers in the selected two positions are exchanged, and this process is repeated $r'$ times.

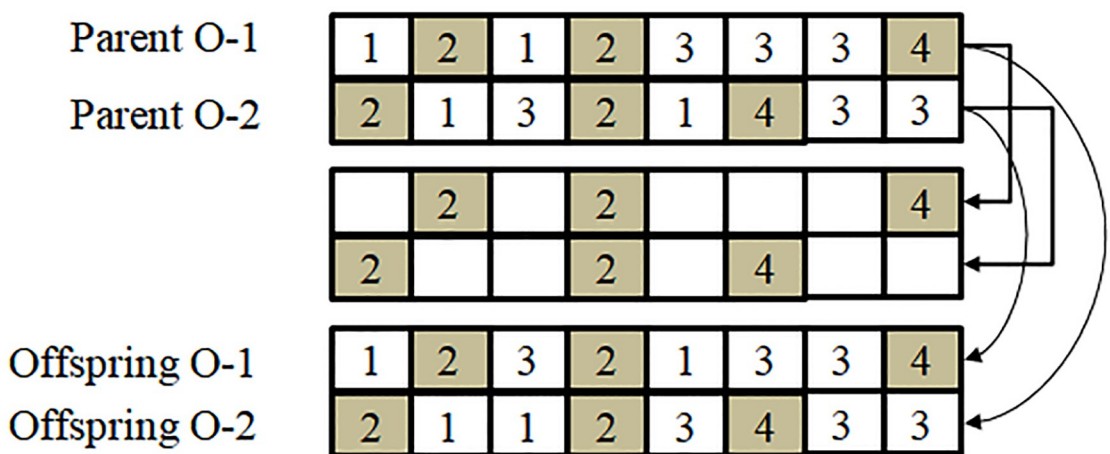

**Fig 4. Schematic diagram of operation crossover.**

## 2.4 Evaluation index

To explore the influencing mechanisms of the propagation rate of advantageous genes on the performance of MPGA-ER, it is necessary to know how to measure this rate. Most existing literature measured the propagation rate of advantageous genes based on takeover time [12, 15], and then studied the algorithm's selection pressure from a theoretical perspective. However, the selection pressure of an algorithm is not related to the performance of the algorithm when solving practical problems; therefore, the takeover time proposed in these studies cannot be directly used to measure the propagation rate of advantageous genes in our study. According to the coding characteristics, HDE is proposed to measure the propagation rate of advantageous genes within and between sub-populations. When individuals in a sub-population evolve through the operators of GA, advantageous genes accumulate gradually, and good individuals gradually fill the whole sub-population. Then, the "differences" between these individuals in a sub-population are also reduced over time. Meanwhile, elite individuals migrate between sub-populations; therefore, advantageous genes propagate from one sub-population to another. Then, the "differences" between these elite individuals of different sub-populations are reduced over time. Therefore, the Hamming distance between all elite individuals of different sub-populations can be used to measure their differences in order to measure the propagation rate of the advantageous genes. However, measuring the propagation rate with the average of the Hamming distances of all individual pairs in the elite set is time-consuming. Therefore, in this study, $X$ individual pairs are randomly selected from the elite set to form a sample set, and the sample set mean then replaces the population mean. Consequently, HDE is defined by Eq (12) [17]:

$$\text{HDE} = \left( \sum_{i=1}^{X} \sum_{j=1}^{2 \times J_t} (1 - \delta(h_{ij}^1, h_{ij}^2)) \right) / (X \times 2 \times J_t) \tag{12}$$

where $\delta(.,.)$ is the Kronecker delta. When two independent variables are identical, then $\delta(.,.)$ equals 1; otherwise, 0. $h_{ij}^1$ and $h_{ij}^2$ represent the two integers at the $j^{\text{th}}$ position of the $i^{\text{th}}$ pair of the elite individuals. In this study, $X$ is set to 100.

As described in [4], GA is actually a random search algorithm with some control strategies. The larger the TIN of a GA, the better is the solution found by this GA. In other words, if two algorithms find the same solution, the algorithm with a smaller TIN performs better. Therefore, while comparing the performances of different algorithms, especially while studying the influences of parameters on their performance, a constant, here, TIN, should be considered. Therefore, an average optimal value (AOV) and a success rate (SR) based on TIN are used in this study to measure the performance of MPGA-ER. For a given TIN, every algorithm runs several times (denoted by $N_{tol}$) independently. The AOV is defined as in Eq (13):

$$\text{Average Optimal Value} = \frac{1}{N_{tol}} \sum_{i=1}^{N_{tol}} AOV_i \tag{13}$$

where $AOV_i$ refers to the best solution of the corresponding FJSP problem found by the algorithm in a single run. $N_{tol}$ is 10 in this study. SR is defined in Eq (14):

$$\text{Success Rate} = \frac{N_{suc}}{N_{tol}} \times 100\% \tag{14}$$

where $N_{suc}$ denotes the number of times MPGA-ER can find the optimal value (the optimal value herein refers to the best solution for the corresponding FJSP problem that could be

found in the literature to date) of the corresponding FJSP problem among $N_{tol}$ independent runs.

Finally, the size of the maximum connected sub-graph (SMCS) and average shortest path (ASP) are used to measure the ER network structures. Furthermore, the relationship between the network structure and the propagation rate of advantageous genes is explored. SMCS is defined as the number of nodes of the maximum connected sub-graph in a network [28]. ASP is calculated as shown in Eq (15):

$$\text{Average Shortest Path} = \sum_{i,j=1, i \neq j}^{N} L_{ij} / (N \times (N-1)) \tag{15}$$

where $L_{ij}$ represents the shortest path between nodes $V_i$ and $V_j$; the definition of the shortest path is similar to that used in [28]. When an ER network is not connected (SMCS is smaller than the total number of nodes), ASP is defined to be infinity.

## 3. Simulation study

First, we study how the sub-population size, $S$, and the sub-population number, $N$, affect the performance of MPGA-ER with a certain TIN. Subsequently, we address how the connection probability, $P$, and the migration frequency, $R$, affect the performance of MPGA-ER. Finally, the effectiveness of solving FJSP by MPGA-ER is verified by solving more FJSP examples.

### 3.1 Effect of sub-population size on MPGA-ER

A 10×10 FJSP instance [29] is considered as an example for studying the influence of the sub-population size, $S$, on the performance of MPGA-ER. This instance is a medium-size FJSP and has been widely studied in several studies [4, 17, 29]. This instance is neither too simple nor too difficult, giving us the opportunity to investigate the performance of MPGA-ER under quite different parameters. To connect all sub-populations together, $P$ is set to 0.02, thereby obtaining a connected ER network. To ensure that the migration frequency, $R$, does not affect the results, $R$ is set to a large number, for example, 10,000. In [17], in which they solved the same FJSP instance, the TIN was 1,600,000. To provide more chances for communication among sub-populations, we set TIN to be 2,000,000. The remaining parameters are set as follows: $N = 100$, $P_r = 0.08$, and $S$ changes from 10 to 150 with a step size of 10. Each algorithm runs independently 10 times, and we use the average of the values of these runs to measure the performance of MPGA-ER. Fig 5 shows the simulation results. In Fig 5, the X-axis, Y-axis (left), and Y-axis (right) represent $S$, AOV, and SR, respectively. For $S = 10$, the performance of MPGA-ER is extremely poor for AOV is 10.1. The optimal value of 7 cannot be found at this point (the optimal value of 10×10 FJSP is 7, and Fig 6 shows its Gantt chart). For increasing $S$, the performance of MPGA-ER improved rapidly. For $S = 40$, the optimal value of 7 is found (AOV is 7.9, and SR is 20%). For $S = 50$, MPGA-ER shows an improved performance with an AOV of 7.4, and SR is 60%. When $S$ is 80 or 100, MPGA-ER performs best, and AOV is 7.3, and SR is 70%. However, as $S$ continues to increase, the performance of MPGA-ER slowly decreases; however, the optimal value, 7, can still be found. Therefore, it can be concluded that, under certain TIN, the performance of MPGA-ER improves rapidly with increasing $S$, and then decreases slowly. To obtain an improved MPGA-ER, the value of $S$ cannot be too small; more specifically, a value greater than or equal to 50 is an appropriate choice.

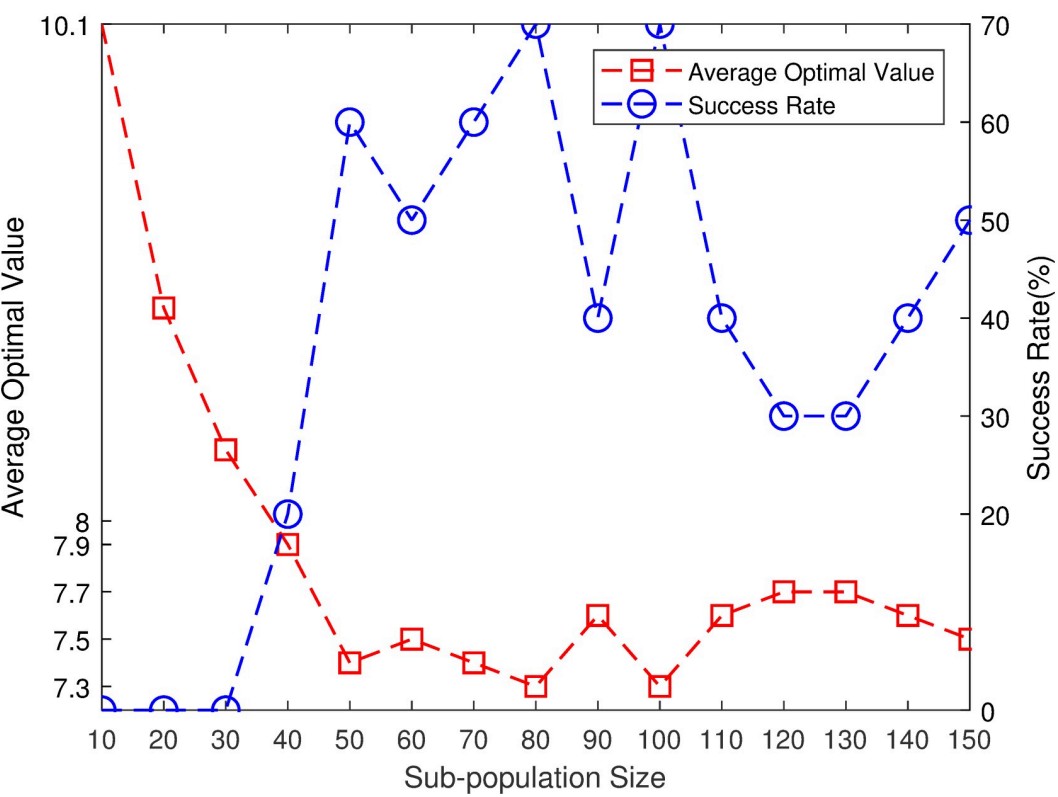

**Fig 5. Effect of sub-population size on MPGA-ER.**

## 3.2 Effect of sub-population number on MPGA-ER

The aforementioned 10×10 FJSP example is considered again. First, we study the influence of the sub-population number, *N*, on the performance of MPGA-ER. *N* is changed from 20 to 200 with a step size of 20, and *S* = 80. The rest of the parameters are similar to those chosen in Section 3.1. Fig 7 presents the simulation results.

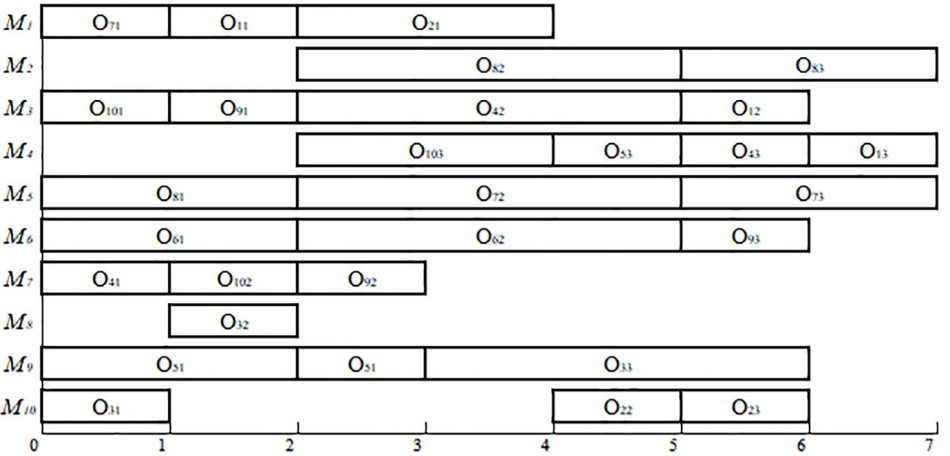

**Fig 6. Gantt chart of the 10×10 FJSP example.**

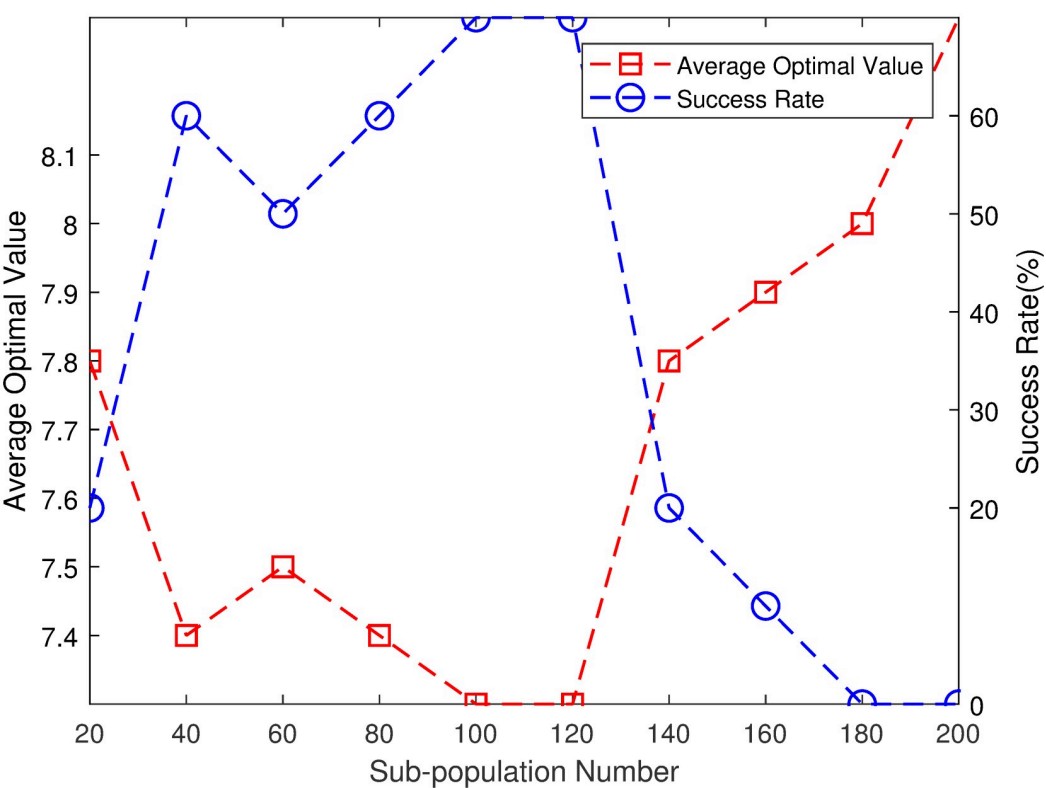

**Fig 7. Effect of sub-population number on MPGA-ER.**

In Fig 7, the X-axis, Y-axis (left), and Y-axis (right) represent $N$, AOV, and SR, respectively. For $N = 20$, the performance of MPGA-ER is poor and AOV is 7.8, and SR is 20%. The performance of MPGA-ER increases slowly with increasing $N$. For $N = 40$, AOV is 7.4, and SR is 60%. For $N$ is 100 or 120, MPGA-ER performs the best, AOV is 7.3, and SR is 70%. As $N$ continues to increase, the performance of MPGA-ER rapidly decreases. For $N = 180$, MPGA-ER cannot find the optimal value of 7. Therefore, it can be concluded that, with certain TIN, the performance of MPGA-ER first improves slowly and then decreases rapidly with increasing $N$. To obtain an improved MPGA-ER, the value of $N$ should not be too small or too large; more specifically, a value between 40 and 120 is an appropriate choice.

### 3.3 Effect of connection probability on MPGA-ER

Considering the aforementioned example of 10×10 FJSP again, we study the influence of the connection probability, $P$, on the performance of MPGA-ER. The related parameters are set as follows: TIN = 2,000,000; $N = 100$; $S = 80$; $P_r = 0.08$; $R = 10,000$; $P = 0, 0.001, 0.002, 0.003, 0.004, 0.005, 0.006, 0.007, 0.008, 0.009, 0.01, 0.02, 0.03, 0.04, 0.05, 0.06, 0.07, 0.08, 0.09, 0.1, 0.2, 0.3, 0.4, 0.5, 0.6, 0.7, 0.8, 0.9, 1$. Each algorithm runs 10 times independently, and we use the average of the optimum values of these runs to measure the performance of MPGA-ER. Table 4 lists the values of AOV, SR, ASP, and SMCS for each $P$ value. Fig 8 illustrates the curves of HDE against the iteration times (IT) for several values of $P$. In Fig 8, the X- and Y-axes represent the values of IT and HDE, respectively. For TIN = 2,000,000, $S = 80$, $N = 100$, then IT is 250. Consequently, each curve in Fig 8 should have 250 points. To clearly express these curves, we consider a point every 10 generations; therefore, there are approximately 26

**Table 4. Values of AOV, SR, ASP, and SMCS for different values of *P*.**

| *P* | AOV | SR (%) | ASP | SMCS |
|---|---|---|---|---|
| 0 | 9.5 | 0 | ∞ | 0 |
| 0.001 | 8.4 | 0 | ∞ | 4 |
| 0.002 | 8.1 | 10 | ∞ | 10 |
| 0.003 | 8 | 10 | ∞ | 12 |
| 0.004 | 7.6 | 40 | ∞ | 22 |
| 0.005 | 7.4 | 70 | ∞ | 30 |
| 0.006 | 7.7 | 40 | ∞ | 65 |
| 0.007 | 7.4 | 60 | ∞ | 72 |
| 0.008 | 7.1 | 90 | ∞ | 75 |
| 0.009 | 7.1 | 90 | ∞ | 83 |
| 0.01 | 7.3 | 70 | ∞ | 91 |
| 0.02 | 7.4 | 60 | 3.4584 | 100 |
| 0.03 | 7.4 | 60 | 2.8527 | 100 |
| 0.04 | 7.2 | 80 | 2.3653 | 100 |
| 0.05 | 7.5 | 50 | 2.2194 | 100 |
| 0.06 | 7.5 | 50 | 2.1077 | 100 |
| 0.07 | 7.8 | 20 | 1.9937 | 100 |
| 0.08 | 7.8 | 20 | 1.9083 | 100 |
| 0.09 | 7.8 | 20 | 1.8591 | 100 |
| 0.1 | 7.9 | 10 | 1.8273 | 100 |
| 0.2 | 8 | 0 | 1.6372 | 100 |
| 0.3 | 8.1 | 0 | 1.4812 | 100 |
| 0.4 | 8 | 0 | 1.3479 | 100 |
| 0.5 | 8.1 | 0 | 1.2501 | 100 |
| 0.6 | 8 | 10 | 1.1591 | 100 |
| 0.7 | 8 | 10 | 1.0883 | 100 |
| 0.8 | 8.1 | 0 | 1.0391 | 100 |
| 0.9 | 8 | 0 | 1.0113 | 100 |
| 1 | 8.2 | 0 | 1 | 100 |

points in each curve. For *P* = 0, the corresponding ER network is a graph that includes only isolated points (at this time, MPGA-ER is equivalent to 100 standard GAs running simultaneously). The advantageous genes propagate only within the sub-populations; therefore, HDE declines very slowly with the evolution of the algorithm. Combined with Table 4, the AOV of this algorithm is 9.5, whereas the optimal value, 7, cannot be found. Meanwhile, the ASP of the corresponding ER network is infinite, and the propagation rate of the advantageous genes is extremely low. As *P* increases, SMCS gradually increases, and advantageous genes begin to propagate among different sub-populations; therefore, the propagation rate becomes higher. For *P* = 0.001, SMCS is 4. There will be some migrations of elite individuals between these four sub-populations; therefore, HDE will decline faster. The AOV of this algorithm is 8.4, which is obviously smaller than 9.5 for *P* = 0; however, the optimal value of 7 cannot be found. Table 4 shows that for *P* = 0.002, MPGA-ER starts to find the optimal value, 7 and AOV is 8.1 and SR is 10%. When *P* = 0.005, SMCS is 30, the propagation rate is larger, and HDE eventually drops to approximately 0.2. However, the ER network is still not connected; therefore, advantageous genes do not fill all sub-populations, and the HDE value does not drop to 0. The AOV of this algorithm is 7.4, and SR is 70%, which indicates that the performance of MPGA-ER improves as the propagation rate increases. For *P* = 0.009, SMCS

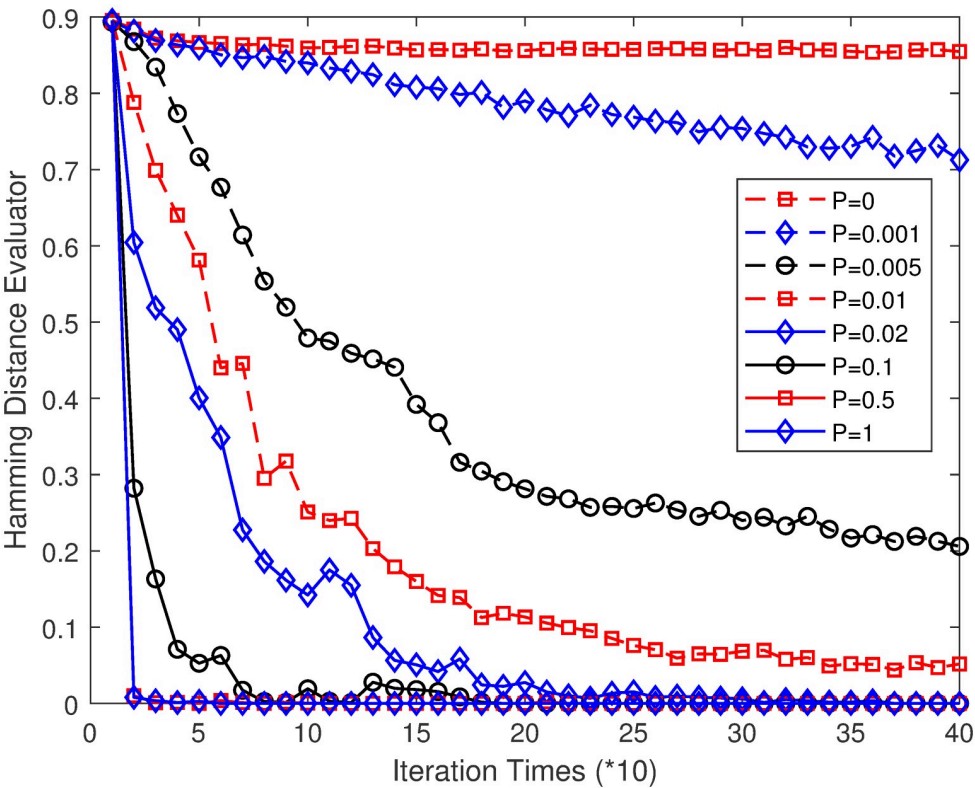

**Fig 8. HDE curves over the iteration times for several values of *P*.**

is 83, for which the ER network is close to the connected network. At this point, the propagation rate of the advantageous genes is faster, and the HDE drops to approximately 0.07. At this time, MPGA-ER performs best for an AOV of 7.1, and SR is 90%. For $P = 0.02$, SMCS is 100, the network is a connected network, and the value of ASP is 3.4584. At this time, the HDE becomes 0 at approximately the 250[th] generation, at which point the advantageous genes fill all sub-populations. Additionally, the algorithm performs satisfactorily for AOV is 7.4, and SR is 60%. As *P* continues to increase, ASP continues to decrease, and the advantageous genes propagate faster over time. For example, for $P = 0.1$, ASP is 1.8273, and HDE drops to 0 at approximately the 100[th] generation. The performance of this algorithm becomes worse whenever SR is only 10%. For $P = 0.5$, ASP is only 1.2501, and HDE drops to 0 at approximately the 50[th] generation. The propagation rate of the advantageous genes is considerably high; the performance of this algorithm is extremely poor because it cannot find the optimal value of 7. Finally, for $P = 1$, the network becomes a complete graph, and each sub-population communicates with all other sub-populations in each generation. Further, the advantageous genes propagate the fastest, and the corresponding algorithm performs very poorly; however, it is still better than the standard GA. Therefore, it can be concluded that as *P* changes from 0 to 1, the network gradually changes from a graph including just isolated points (for $P = 0$) to a connected network (for $P = 0.02$), and further to a complete graph (for $P = 1$). The propagation rate of advantageous genes increases over time; however, the performance of MPGA-ER first increases rapidly, and then begins to decrease slowly. To obtain an improved MPGA-ER, the value of *P* should be greater than 0 but considerably less than 1; more specifically, a value between 0.004 and 0.04 is an appropriate choice.

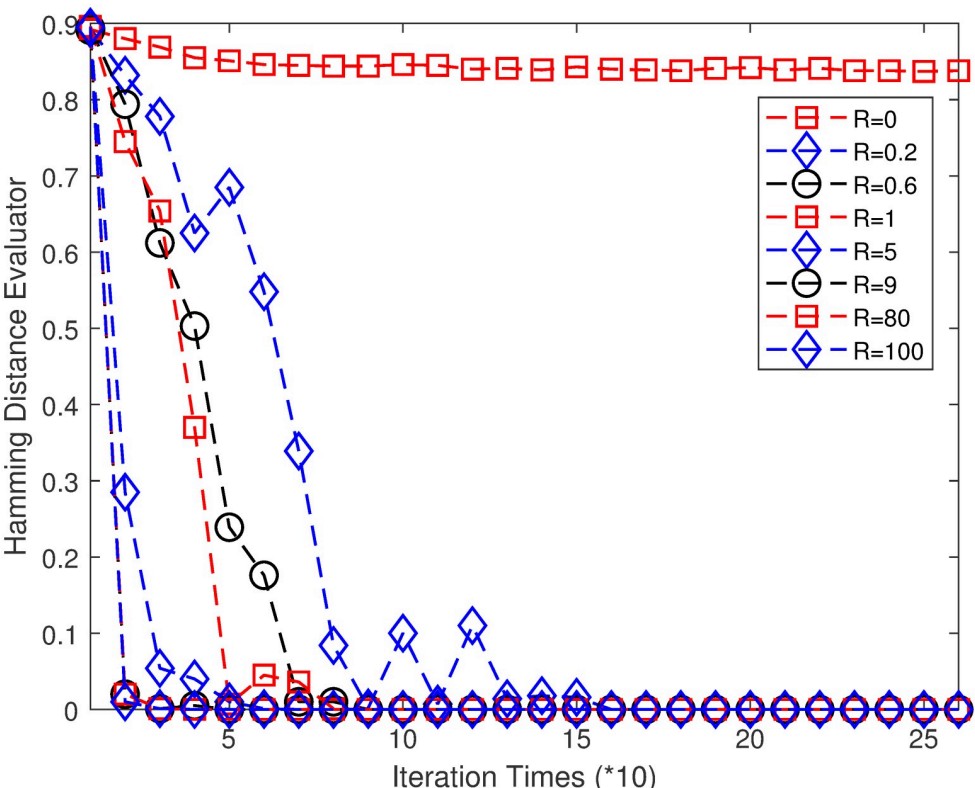

**Fig 9. HDE curves over the iteration times for several values of R.**

## 3.4 Effect of migration frequency on MPGA-ER

The aforementioned example of a 10×10 FJSP is considered again, and the influence of $R$ on the performance of MPGA-ER is studied. To eliminate the effect of $P$ on the results in this subsection, $P$ is set to 1. The other parameters are set as follows: TIN = 2,000,000; $N$ = 100; $S$ = 80; $P_r$ = 0.08; $R$ = 0, 0.2, 0.4, 0.6, 0.8, 1, 3, 5, 7, 9, 20, 40, 60, 80, 100. Every algorithm runs 10 times independently. Fig 9 illustrates the HDE curves over the ITs for several values of $P$. Table 5 lists the values of AOV, SR, and the communication times (CT) for different values of $P$.

The data points in Fig 9 are selected to be similar to those in Fig 8, thereby resulting in 26 points. For $R$ = 0, the condition is similar to when $P$ = 0 in Fig 8. Here, no elite individuals propagate between different sub-populations; therefore, the HDE values remain almost unchanged (as shown in Fig 9). As $R$ increases, the CT increases, and the propagation rate accelerates. For $R$ = 0.2, CT is 41. HDE decreases rapidly with increasing IT; it drops to 0 at approximately the 140th generation. Combined with Table 5, the algorithm can find the optimal value of 7, thereby achieving 30% success. For $R$ = 0.6, CT is 93, and the advantageous genes propagate faster; therefore, HDE decreases to 0 at approximately the 70th generation.

**Table 5. Values of AOV, SR, and CT for different values of R.**

| R | 0 | 0.2 | 0.4 | 0.6 | 0.8 | 1 | 3 | 5 | 7 | 9 | 20 | 40 | 60 | 80 | 100 |
|---|---|---|---|---|---|---|---|---|---|---|---|---|---|---|---|
| AOV | 9.3 | 7.8 | 7.6 | 7.4 | 7.5 | 7.6 | 7.4 | 7.5 | 7.4 | 7.7 | 8 | 8 | 8.1 | 8.2 | 8.2 |
| SR (%) | 0 | 30 | 50 | 60 | 50 | 40 | 60 | 50 | 60 | 30 | 10 | 10 | 0 | 0 | 0 |
| CT | 0 | 41 | 75 | 93 | 112 | 126 | 187 | 210 | 217 | 225 | 238 | 244 | 246 | 247 | 248 |

However, the algorithm can still find the optimal value of 7 and achieves a 60% success. For $R = 9$, CT is 217, and the propagation rate of the advantageous genes is extremely high; therefore, HDE decreases to 0 at approximately the 30th generation, which is similar to the propagation rate for $P = 0.5$ (it decreases to 0 at approximately the 40th generation). The algorithm still performs satisfactorily in this case and achieves 60% success. However, for $P = 0.5$ shown in Table 4, the algorithm shows extremely poor performance, and the optimal value cannot be found. This means that the performance of MPGA-ER with a tunable communication strategy, that is, less communication at an earlier stage and more communication at a later stage is better than that of MPGA-ER with an invariant communication strategy. For $R$ greater than 20, the performance of MPGA-ER becomes extremely poor, which is similar to that of MPGA-ER with an invariant communication strategy. Therefore, it can be concluded that a tunable communication strategy with less communication at an earlier stage and more communication at a later stage is an effective strategy for improving the performance of MPGA-ER. Here, a value of $R$ between 0.6 and 7 is an appropriate choice.

## 3.5 Effectiveness of MPGA-ER in solving FJSP

To verify the effectiveness of MPGA-ER, it is employed to solve more of the FJSP instances mentioned in [30], and is compared with that of other algorithms (AIA and HHS in [31], M2 in [32], MILP in [33], and HA in [34]). These instances, including small-size FJSP (SFJS01 – SFJS10) and medium- and large-size FJSP (MFJS01 –MFJS08), have been widely used in previous studies [30–34]; hence, it is convenient to solve them by using MPGA-ER and then compare it with results obtained using other algorithms directly. These algorithms were chosen due to their simplicity and wide applicability. In line with previous researchers, we use the optimal value found by an algorithm to measure its performance. The optimal values found by other algorithms are cited directly from the relevant papers. According to Sections 3.1–3.4, the related parameters of MPGA-ER are set as follows: $P_r = 0.08$, $P = 0.009$, $S = 100$, $N = 80$, $R = 3$, and IT = 1,000. Table 6 lists the results of the comparison.

As shown in Table 6, when solving MFJS02 using MPGA-ER, the optimal value is 446, which is better than the value of 448 found by the AIA algorithm. When solving MFJS05 using MPGA-ER, the optimal value is 514, which is better than the value of 527 that was found by the AIA algorithm. When solving MFJS06 using MPGA-ER, the optimal value is 634, which is better than the value of 635 that was found by the AIA algorithm. While solving MFJS04 using MPGA-ER, the optimal value is 554, which is better than the value of 564 found by the algorithm M2. When solving MFJS07 using MPGA-ER, the optimal value is 879, which is better than the value of 928 that was found by algorithm M2. Meanwhile, the optimal values of all 18 FJSP instances can be found with MPGA-ER. Therefore, the effectiveness of MPGA-ER could be verified. Fig 10 illustrates the Gantt chart of MFJS08.

## 4. Analysis of results and discussion

As described in Section 3.1, for a certain TIN, the performance of MPGA-ER first increases rapidly and then decreases slowly with increasing sub-population size. When $S$ is small, a sub-population consists of few individuals; therefore, the gene pool of the small sub-population is naturally uniform. Consequently, the performance of the corresponding MPGA-ER is considerably poor. As $S$ increases, there are enough individuals in a single sub-subpopulation for the maintenance of population diversity; therefore, the performance of the corresponding MPGA-ER improves. This implies that when an algorithm engineer wants to design an improved multi-population GA, the sub-population size cannot be too small.

**Table 6. Results of the 18 FJSP instances.**

| Problems | AIA and HHS [31] | | M2 [32] | MILP [33] | HA [34] | MPGA-ER |
|---|---|---|---|---|---|---|
| | Optimal Value (AIA) | Optimal Value (HHS) | Optimal Value | Optimal Value | Optimal Value | Optimal Value |
| SFJS01 | 66 | 66 | 66 | 66 | 66 | 66 |
| SFJS02 | 107 | 107 | 107 | 107 | 107 | 107 |
| SFJS03 | 221 | 221 | 221 | 221 | 221 | 221 |
| SFJS04 | 355 | 355 | 355 | 355 | 355 | 355 |
| SFJS05 | 119 | 119 | 119 | 119 | 119 | 119 |
| SFJS06 | 320 | 320 | 320 | 320 | 320 | 320 |
| SFJS07 | 397 | 397 | 397 | 397 | 397 | 397 |
| SFJS08 | 253 | 253 | 253 | 253 | 253 | 253 |
| SFJS09 | 210 | 210 | 210 | 210 | 210 | 210 |
| SFJS10 | 516 | 516 | 516 | 516 | 516 | 516 |
| MFJS01 | 468 | 468 | 468 | 468 | 468 | 468 |
| MFJS02 | 448 | 446 | 446 | 446 | 446 | 446 |
| MFJS03 | 468 | 466 | 466 | 466 | 466 | 466 |
| MFJS04 | 554 | 554 | 564 | 554 | 554 | 554 |
| MFJS05 | 527 | 514 | 514 | 514 | 514 | 514 |
| MFJS06 | 635 | 634 | 634 | 634 | 634 | 634 |
| MFJS07 | 879 | 879 | 928 | 879 | 879 | 879 |
| MFJS08 | 884 | 884 | /[a] | /[a] | 884 | 884 |

[a]Symbol "/" indicates that the corresponding value is not provided in the corresponding paper.

As described in Section 3.2, for a certain TIN, the performance of MPGA-ER first increases slowly, and then decreases rapidly with increasing sub-population number. A small value of $N$ is not conducive to maintaining population diversity; however, the number of individuals in a single sub-population is moderate (the number of individuals in a single sub-population in Section 3.2 is 80), which is conducive to maintaining gene diversity. Therefore, the performance of the corresponding MPGA-ER is not very poor. With increasing $N$, the number of individuals used in an iteration is very large (for $N = 200$, the number of individuals used in an iteration is 16,000); therefore, the total number of iterations is very small (for $N = 200$, the number of iterations is only 125). Individuals do not have enough time to accumulate advantageous genes. Therefore, the performance of the corresponding MPGA-ER worsens. This implies that $N$ should not be too small or too large when an algorithm engineer wants to design an improved multi-population GA.

As described in Section 3.3, for $P = 0$, MPGA-ER is equivalent to 100 standard GAs running simultaneously. The main operator of the GA is a crossover that readily accumulates advantageous genes without being destroyed, thereby leading to premature convergence.

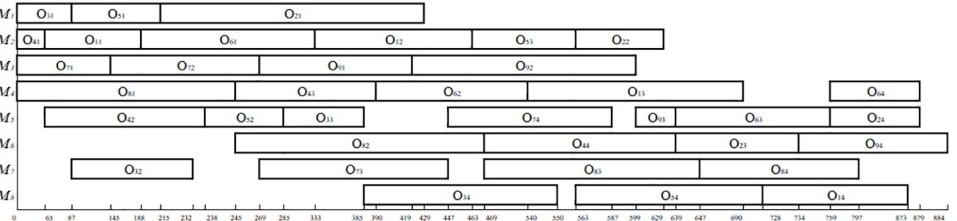

**Fig 10. Gantt chart of MFJS08.**

The performance of the corresponding MPGA-ER is very poor. As $P$ increases, the subgraph of the network gradually becomes larger; therefore, advantageous genes propagate among different sub-populations, thereby causing a larger propagation rate. Individuals of different sub-populations can use the advantageous genes of the other sub-populations to improve their fitness; therefore, the performance of the corresponding MPGA-ER gradually improves. For $P = 0.008$, MPGA-ER performs the best; however, the network is still not a connected network. When $P$ is greater than 0.02, the network starts to become a connected network. At this time, individuals in a sub-population have the opportunity to communicate with all other sub-populations to improve their fitness; therefore, the performance of MPGA-ER is still considerably good. However, as $P$ continues to increase, the neighboring sub-populations of a sub-population also increase. In each communication, the advantageous genes of a sub-population propagate to other sub-populations; therefore, the propagation rate increases over time, which leads to the advantageous genes filling all sub-populations more quickly. Therefore, the corresponding MPGA-ER is similar to the standard GA, thereby causing advantageous genes to accumulate rapidly without being destroyed. Therefore, the performance of MPGA-ER starts to decrease again. As shown in Table 4, the performance of MPGA-ER for $P = 0$ (equivalent to 100 standard GAs running simultaneously) is not better than that of MPGA-ER for $P = 1$, let alone that of MPGA-ER when $P = 0.009$, which is the best, implying that its performance shows considerable improvement compared with traditional GA. This implies that to obtain an improved multi-population GA, an appropriate value of $P$ should be adopted to ensure that the propagation rate is neither so slow that individuals cannot benefit from other sub-populations nor too fast so that a single elite individual quickly fills all sub-populations. Additionally, in previous studies (such as [9, 15]), to obtain a limited takeover time, all networks involved were connected. However, according to our simulation results, to obtain an improved MPGA-ER, network connectivity is not necessary. For example, the performance of MPGA-ER for $P = 0.009$ (the corresponding ER network is not connected completely) is better than that of MPGA-ER for $P = 0.02$ (the corresponding ER network is connected completely). When $P = 0.009$, the maximum connected sub-graph of the corresponding ER network was 83, and the advantageous genes could then be propagated among these sub-populations. However, there are still 17 sub-populations that are not connected with the maximum connected sub-graph; hence, the advantageous genes of these 17 sub-populations cannot be communicated with the sub-populations in the maximum connected sub-graph. This means that the advantageous genes can be propagated among most sub-populations; however, they cannot be propagated among a small number of sub-populations, which ensures that the propagation rate is neither so slow that individuals can benefit from other sub-populations (among 83 sub-populations) or too fast so that a single elite individual quickly fills all sub-populations (cannot fill the other 17 sub-populations). This implies that network connectivity is not necessary when an algorithm engineer wants to obtain an improved MPGA-ER.

As described in Section 3.4, a tunable communication strategy with less communication in the earlier stages and more communication in later stages is an effective strategy. Less communication in the earlier stages is beneficial for accumulating advantageous genes within a subpopulation, whereas more communication in the later stages is beneficial for accumulating advantageous genes between different sub-populations and promoting algorithm convergence. This implies that the performance of the corresponding MPGA-ER with the aforementioned strategy is better than that of the MPGA-ER with an invariant communication strategy under the same conditions. This may shed light on how an algorithm engineer should design an improved multi-population GA when considering variable migration frequency.

## 5. Conclusion and future work

In this study, an ER network with different connection probabilities and variable migration frequency was used to control the propagation rate of advantageous genes between sub-populations of a GA, thereby obtaining an MPGA-ER. Using MPGA-ER to solve an FJSP instance, the influencing mechanisms of the propagation rate of advantageous genes on the performance of MPGA-ER were addressed. Meanwhile, the influencing mechanisms of parameters $N$ and $S$ on MPGA-ER under a certain TIN were also studied. The simulation results indicate that MPGA-ER shows considerable performance improvement compared with the standard GA. The propagation rate of advantageous genes has a significant impact on the performance of MPGA-ER—when the propagation rate is extremely low or extremely high, the performance of MPGA-ER is poor. Only a moderate propagation rate can ensure a satisfactory performance of the algorithm. Specifically, the desired intervals for $P$ and $R$ are between [0.004, 0.04] and [0.6, 7], respectively. Unlike shown in previous studies, to obtain an improved MPGA-ER, network connectivity is not necessary. Additionally, parameters $N$ and $S$ have a significant impact on the performance of MPGA-ER—for a certain TIN, the performance of MPGA-ER first increases slowly, and then decreases rapidly with increasing $N$. More specifically, the desired interval of $N$ is [40, 120]. The performance of MPGA-ER first increases rapidly and then decreases slowly with increasing $S$; more specifically, the desired value of $S$ is greater than or equal to 50. Our results may shed light on how to use an ER network to achieve a better MPGA.

In this study, we used a multi-population to improve a standard GA for obtaining a MPGA-ER, which may limit its performance. As described in [4], there are five main methods (multi-population is just one of them) to improve GA and other algorithms. In our future work, we will first use another method to improve the standard GA or other algorithms, and then, use ER or other networks to design improved algorithms, thereby obtaining a corresponding multi-population algorithm that may perform even better.

## Supporting information

**S1 File.**
(RAR)

## Author Contributions

**Conceptualization:** Xiaoqiu Shi, Wei Long, Yanyan Li.

**Data curation:** Xiaoqiu Shi.

**Methodology:** Yanyan Li.

**Resources:** Yanyan Li, Dingshan Deng.

**Software:** Xiaoqiu Shi, Dingshan Deng.

**Supervision:** Wei Long, Yanyan Li.

**Writing – original draft:** Xiaoqiu Shi.

**Writing – review & editing:** Xiaoqiu Shi, Yanyan Li, Dingshan Deng.

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
