## [Decision Letter · Decision Letter 0]

26 Feb 2020

PONE-D-19-22005

Multi-population genetic algorithm with ER network to solve flexible job shop scheduling problem

PLOS ONE

Dear Dr. li,

Thank you for submitting your manuscript to PLOS ONE. After careful consideration, we feel that it has merit but does not fully meet PLOS ONE’s publication criteria as it currently stands. Therefore, we invite you to submit a revised version of the manuscript that addresses the points raised during the review process.

Please respond to the reviewers' comments carefully and make the requested amendments accordingly. In addition, please highlight the amended parts in your paper which help the reviewers to find the corrections easier. 

We would appreciate receiving your revised manuscript by Apr 11 2020 11:59PM. To enhance the reproducibility of your results, we recommend that if applicable you deposit your laboratory protocols in protocols.io, where a protocol can be assigned its own identifier (DOI) such that it can be cited independently in the future. For instructions see: http://journals.plos.org/plosone/s/submission-guidelines#loc-laboratory-protocols

We look forward to receiving your revised manuscript.

Kind regards,

Ashkan Memari

Academic Editor

PLOS ONE

Journal Requirements:

Reviewers' comments:

Reviewer's Responses to Questions

**Comments to the Author**

1. Is the manuscript technically sound, and do the data support the conclusions?

Reviewer #1: Yes

Reviewer #2: Partly

2. Has the statistical analysis been performed appropriately and rigorously? 

Reviewer #1: Yes

Reviewer #2: Yes

3. Have the authors made all data underlying the findings in their manuscript fully available?

Reviewer #1: No

Reviewer #2: Yes

4. Is the manuscript presented in an intelligible fashion and written in standard English?

Reviewer #1: Yes

Reviewer #2: No

5. Review Comments to the Author

Reviewer #1: Title: Multi-population genetic algorithm with ER network to solve flexible job shop scheduling problem

In the study “Multi-population genetic algorithm with ER network to solve flexible job shop scheduling problem”, the authors propose a multi-population GA with ER network to solve the flexible job shop scheduling problem. They can show that their developed algorithm can dominate the traditional GA.

This topic is interesting. However, a major revision is needed to improve the paper quality. Some important comments are given below:

1. The writing and English should be improved. Pay careful attention to the usage of articles. Moreover, you can’t use “research” as a countable noun.

2. Number the sections and subsections for better positioning.

3. This topic has a widespread background. So, I suggest the authors use a tabular form for better comparisons.

4. Add the domain of variables to the mathematical model of FJSP.

5. My major concern is regarding the novelty of the algorithm, which combines a number of methods. Two questions are raised to me. 1) Why these methods are selected rather than others? 2) What are the specialties for the studied problem?

6. How could you set the parameters of the algorithm?

7. You should explain the main limitations of your proposed algorithm. Moreover, the practical aspect of the problem you studied should be discussed in more detail.

8. The authors may read and consider the following references to improve the introduction of the manuscript.

- https://doi.org/10.1016/j.cie.2019.106007

- https://doi.org/10.1080/21693277.2019.1620651

- https://doi.org/10.3390/su10051366

Reviewer #2: Most of this work is already published in previous work, the methodology should be clearly elaborated, used terminology must be explained, and findings should be reviewed and proved.

The comments to the manuscripts are available in the attached word file, Please refer to

6. PLOS authors have the option to publish the peer review history of their article (what does this mean?). If published, this will include your full peer review and any attached files.

Reviewer #1: Yes: Erfan Babaee Tirkolaee

Reviewer #2: Yes: Moath Alatefi and Mustafa Saleh

---

## [Author Response · Author response to Decision Letter 0]

23 Mar 2020

The main changes in the manuscript and the responds to the reviewers’ comments are as flows. 

Reviewer #1

In the study “Multi-population genetic algorithm with ER network to solve flexible job shop scheduling problem”, the authors propose a multi-population GA with ER network to solve the flexible job shop scheduling problem. They can show that their developed algorithm can dominate the traditional GA.

This topic is interesting. However, a major revision is needed to improve the paper quality. Some important comments are given below:

1. The writing and English should be improved. Pay careful attention to the usage of articles. Moreover, you can’t use “research” as a countable noun.

Responses: According to your comments, we have improved our writing and English language carefully by us. We also ask “Editage” for a professional English language editing.

2. Number the sections and subsections for better positioning.

Responses: According to your comments, we have numbered the sections and subsections.

3. This topic has a widespread background. So, I suggest the authors use a tabular form for better comparisons.

Responses: According to your comments, we have used a Table to compare several evolutionary algorithms with different networks. (see Table 1 in the manuscript)

4. Add the domain of variables to the mathematical model of FJSP.

Responses: According to your comments, we have added the domains of all variables to our mathematical model. (see Equations (1)~(10))

5. My major concern is regarding the novelty of the algorithm, which combines a number of methods. Two questions are raised to me. 1) Why these methods are selected rather than others? 2) What are the specialties for the studied problem?

Responses: To avoid premature convergence which is the main disadvantage of the standard genetic algorithm (GA), we used the multi-population method and ER network to improve GA, obtaining the multi-population GA with ER network (MPGA-ER) in this study. We think there are several reasons why we choose GA that is improved by multi-population method: 1) the GA is simplicity, easy realization, and wide applicability; 2) in our previous research [4], we have found that the performance of GA can be improved by using multi-population method solely; and 3) we are familiar with genetic algorithm. There are some reasons why we choose ER network to design multi-population GA (MPGA) as follows. In our previous research [17], we have used seven networks including ER network to design MPGA to address how different networks affect the performance of MPGA, where the connection probability (P) of the ER model was a constant. We found that the propagation rate of advantageous genes among sub-populations was limited by these network topologies; hence, how the propagation rate over a wide range affects the performance of MPGA is still not clear. Fortunately, when the connection probability of the ER model changes from 0 to 1, the corresponding network gradually transforms from a graph that completely includes isolated points to a complete graph; therefore, the propagation rate of the advantageous genes changes from considerably low to high. Accordingly, to know how the propagation rate over a wide range affects the performance of MPGA, we used the multi-population method and ER network to improve GA. We think this may be the specialties of our study. According to your comments, these reasons are described in our revised manuscript directly or indirectly. (see the words in red color in Section 1 of the version named “Revised Manuscript with Track Changes”)

6. How could you set the parameters of the algorithm?

Responses: We set the parameters of the algorithm according to our simulations.

7. You should explain the main limitations of your proposed algorithm. Moreover, the practical aspect of the problem you studied should be discussed in more detail.

Responses: According to your comments, we have explained the main limitations of our algorithm in the Conclusion Section. We also discussed some practical applications of our study in the Conclusion Section.

8. The authors may read and consider the following references to improve the introduction of the manuscript.

- https://doi.org/10.1016/j.cie.2019.106007

- https://doi.org/10.1080/21693277.2019.1620651

- https://doi.org/10.3390/su10051366

Responses: According to your comments, we have read all the three papers you give us carefully to improve our introduction. (see the words in red color in Sections 1 and 2 of the version named “Revised Manuscript with Track Changes”)

Finally, thank you very much, and your comments are constructive and helpful.

Reviewer #2

I want to thank the author for submitting the paper “Multi-population genetic algorithm with ER network to solve flexible job shop scheduling problem”. Even though I find the research idea and the topic in general quite interesting and relevant, I have many concerns that should be addressed:

1. Most of this work is already published in previous work (Shi, Long et al. 2020). As an extension, the authors solved more FJSP instances and compared the obtained results with previous works for the same instances. The authors should clearly explain the similarity and differences between this research and the one mentioned above.

Responses: In our previous research [17] that you mentioned, we have used seven different networks including ER network to design MPGA (multi-population genetic algorithm), thereby addressing how different networks affect the performance of MPGA. However, in [17], the connection probability (P) of the ER model was a constant. We found that the propagation rate of advantageous genes among sub-populations was limited by these network topologies; hence, how the propagation rate over a wide range affects the performance of MPGA is still not clear. Fortunately, when the connection probability of the ER model changes from 0 to 1, the corresponding network gradually transforms from a graph that completely includes isolated points to a complete graph; therefore, the propagation rate of the advantageous genes changes from considerably low to high. Accordingly, as an extension of [17], to know how the propagation rate over a wide range affects the performance of MPGA, we used the ER network to improve GA. These reasons are added to our revised manuscript directly or indirectly. (see the words in red color in Section 1 of the version named “Revised Manuscript with Track Changes”)

2. The methodology of the study should be clearly elaborated, in order to argue your decisions and to remove the doubts.

Responses: According to your comments, the methodology of this study has been read and the revised carefully, such as the description of ER model. (see the words in red color in Section 2 of the version named “Revised Manuscript with Track Changes”)

3. In lines 15-25, the authors stated that they study the effects of sub-population size and number and the propagation rate of advantageous genes on the performance of MPGA-ER. However, they did not define what the performance is.

Responses: According to your comments, we have defined what the performance is in the Abstract Section. (see the words in red color in Abstract Section of the version named “Revised Manuscript with Track Changes”)

4. The language structure needs to be improved.

Responses: According to your comments, we have improved our English language carefully by us. We also ask “Editage” for a professional English language editing.

5. Sections are not numbered.

Responses: According to your comments, we have numbered the Sections and Sub-sections in our revised manuscript.

6. In lines 17 -18, the author mentioned that “the performance shows considerable improvement compared to the traditional GA”. However, this comparison is not found in the work.

Responses: This results is implied as follows: as shown in Table 3, the performance of MPGA-ER when P = 0 (equivalent to 100 standard GAs running simultaneously) is not better than that of MPGA-ER when P = 1, let alone that of MPGA-ER when P = 0.009 which is the best, implying that the performance shows considerable improvement compared to the traditional GA. These descriptions have been added in our revised manuscript. (see the words in red color in Section 4 of the version named “Revised Manuscript with Track Changes”)

7. The study mentioned ER network as the main contribution, however, it is not defined clearly.

Responses: According to your comments, we have defined the ER model more clearly. (see the words in red color in Section 2.2 of the version named “Revised Manuscript with Track Changes”)

8. In line 277, it was mentioned: “The average of the obtained optimal values termed as AOV is used to evaluate the performance of MPGA-ER”. The authors did not clearly define the optimal values.

Responses: Actually, there are two concepts of optimal values in our study, which are not defined clearly. We feel sorry about it. The optimal value in the concept of AOV refers to the best solution of the corresponding FJSP problem found by the algorithm in a single run. In addition, we have added an equation to define AOV (Equation (13)). On the other hand, the optimal value in the concept of SR refers to the best solution of the corresponding FJSP problem can be found so far in the literature. And these descriptions are added to our revised manuscript. (see the words in red color in Section 2.4 of the version named “Revised Manuscript with Track Changes”)

9. In line 280 an equation of success rate was introduced “eq. 7”, “where Nsuc denotes the number of times MPGA-ER can find the optimal value of the corresponding FJSP problem among Ntol independent runs” how the authors guarantee that the solution is optimal?

Responses: Here, the optimal value refers to the best solution of the corresponding FJSP problem can be found so far in the literature. Actually, the optimal value means the best solution of the corresponding FJSP problem can be found so far in the literature, which we defined it as optimal value. 

10. What is the “average optimal value”?

Responses: According to your comments, we have added an equation to define AOV (Equation (13)). 

11. In line 425, it was mentioned that “According to Sections 3.1~3.4”. These numbers were not presented.

Responses: According to your comments, we have numbered all the Sections.

12. In line 426, the reported optimization parameters of MPGA-ER are Pr = 0.08, P = 0.009, S = 100, N = 80, R = 3, and IT = 1000, however various GA parameters have not optimized.

Responses: The parameters Pr and IT are indeed ignored in our study. The parameter Pr of GA is used to increase population diversity, but cannot be set too large. In this study, we mainly use multi-population to increase the diversity of GA. Thus, the parameter Pr is set as a small value randomly in this study. When considering TIN, the parameter IT can be calculated approximately when other parameters are given. In our previous research [4], we addressed how different TINs (associated with IT) affect the performance of MPGA. We found that it cannot affect the performance of MPGA when it is large (larger than 450,000 according to [4]). When it is very small, with increase of TIN, the performance of MPGA becomes better. Thus, the parameter IT is set as a large number randomly in this study. 

13. In Table 4, the authors compared the results with previous works only in terms of “optimal values” (Cmax). However, the CPU times were not presented.

Responses: For fairly, we did not realize these algorithms in other papers by us and cited the results directly from the original papers. Thus, the CPU times may become meaningless and were not given.

14. In Table 4, the authors compared the results with previous works in terms of “optimal values”. Referring to one of these works, [31], these results are the best-obtained solutions (but not proofed to be optimal) as shown below. 

Responses: The optimal value here also refers to the best solution of the corresponding FJSP problem can be found so far in the literature.

15. In the conclusion section (line 506), it was mentioned: “Unlike previous studies, to obtain an improved MPGA-ER, network connectivity is not necessary”. However, this conclusion was not clearly explained in the results section.

Responses: According to your comments, we have explained this conclusion carefully in the Section 4. (see the words in red color in Section 4 of the version named “Revised Manuscript with Track Changes”)

Finally, thank you very much, and your comments are constructive and helpful.

---

## [Decision Letter · Decision Letter 1]

16 Apr 2020

PONE-D-19-22005R1

Multi-population genetic algorithm with ER network for solving flexible job shop scheduling problems

PLOS ONE

Dear Dr. li,

Thank you for submitting your manuscript to PLOS ONE. After careful consideration, we feel that it has merit but does not fully meet PLOS ONE’s publication criteria as it currently stands. Therefore, we invite you to submit a revised version of the manuscript that addresses the points raised during the review process.

There are still a few concerns that have been arisen by the reviewers in your manuscript. Please address the requested amendments carefully and highlight the new changes in your paper.

We would appreciate receiving your revised manuscript by May 31 2020 11:59PM. To enhance the reproducibility of your results, we recommend that if applicable you deposit your laboratory protocols in protocols.io, where a protocol can be assigned its own identifier (DOI) such that it can be cited independently in the future. For instructions see: http://journals.plos.org/plosone/s/submission-guidelines#loc-laboratory-protocols

We look forward to receiving your revised manuscript.

Kind regards,

Ashkan Memari

Academic Editor

PLOS ONE

Reviewers' comments:

Reviewer's Responses to Questions

**Comments to the Author**

1. If the authors have adequately addressed your comments raised in a previous round of review and you feel that this manuscript is now acceptable for publication, you may indicate that here to bypass the “Comments to the Author” section, enter your conflict of interest statement in the “Confidential to Editor” section, and submit your "Accept" recommendation.

Reviewer #1: All comments have been addressed

Reviewer #2: All comments have been addressed

2. Is the manuscript technically sound, and do the data support the conclusions?

Reviewer #1: Yes

Reviewer #2: Yes

3. Has the statistical analysis been performed appropriately and rigorously? 

Reviewer #1: Yes

Reviewer #2: Yes

4. Have the authors made all data underlying the findings in their manuscript fully available?

Reviewer #1: Yes

Reviewer #2: Yes

5. Is the manuscript presented in an intelligible fashion and written in standard English?

Reviewer #1: Yes

Reviewer #2: No

6. Review Comments to the Author

Reviewer #1: Good work. The authors tried to improve the quality of the manuscript based on my previous comments. However, there are still two minor concerns to be resolved.

1- I asked the authors "How could you set the parameters of the algorithm?" I mean what was the motivation? How can you ensure these values are logical?

2- Develop the managerial insights section and provide more useful suggestion based on the main achievements of the study.

Reviewer #2: The authors addresses all the previous comments, however, still there is a need for explanation, and there are some other comments. will be attached

7. PLOS authors have the option to publish the peer review history of their article (what does this mean?). If published, this will include your full peer review and any attached files.

Reviewer #1: No

Reviewer #2: Yes: Moath Alatefi & Mustafa Saleh

---

## [Author Response · Author response to Decision Letter 1]

7 May 2020

Replies to the comments

Dear Editors and Reviewers,

We would like to express our sincere gratitude for all of you. 

Thank you for your letter and for the reviewers’ comments concerning our manuscript entitled “Multi-population genetic algorithm with ER network for solving flexible job shop scheduling problems.” We have studied comments carefully and made some changes. The main changes in the manuscript and the responds to the reviewers’ comments are as flows.

Reviewer #1

Good work. The authors tried to improve the quality of the manuscript based on my previous comments. However, there are still two minor concerns to be resolved.

1. I asked the authors "How could you set the parameters of the algorithm?" I mean what was the motivation? How can you ensure these values are logical?

Responses: We are very sorry to misunderstand you last time. And we are trying to answer your question correctly here. In our study, all parameters of the algorithm are as follows: TIN (total individual number), Pr (mutation probability), P (connection probability of the ER model), S (sub-population size), N (sub-population number), R (a parameter used to control the migration frequency), and IT (iteration times). How the parameters P, S, N, and R affect the performance of the algorithm is our theme in this study and hence they are studied by using simulation. And we think these parameters are set reasonably. The parameters Pr, TIN and IT are indeed ignored in our study. The parameter Pr of GA is used to increase population diversity, but cannot be set too large. In our study, we mainly use multi-population to increase the diversity of GA. Thus, the parameter Pr is set as a small value (0.08) randomly in our study. The parameter TIN was studied in our previous research [4]. We found that it cannot affect the performance of MPGA when it is very large (larger than 450,000 according to [4] for the same FJSP instance). When it is very small, with increase of TIN, the performance of MPGA becomes better. Thus, it is ignored and set to be a large number. In [17], in which they solved the same FJSP instance, the TIN was 1,600,000. To provide more chances for communication among sub-populations, we set TIN to be 2,000,000 in this study randomly. The parameter IT (associated with TIN) is also ignored. And when it is very large it cannot affect the performance of MPGA-ER. Thus, it is also ignored and set to be a large number (1,000) randomly in Section 3.5. According to your comments, these reasons are added to the revised manuscript directly or indirectly.

2. Develop the managerial insights section and provide more useful suggestion based on the main achievements of the study.

Responses: According to your comments, we have tried to improve Section 4 whose title is changed from “Analysis of results” to “Analysis of results and discussion”, giving more discussion and useful suggestion in this section. (see the words in red color in Section 4 of the version named “Revised Manuscript with Track Changes”)

Finally, thank you very much, and your comments are constructive and helpful.

Reviewer #2

The authors address all the previous comments, however, still there is a need for explaination, and there are some other comments.

1. In abstract: “Finally, we use a parameter-optimized MPGA-ER to solve for more FJSP instances and demonstrate its effectiveness by comparing it with other algorithms.” Did the authors propose different algorithms? Besides, In Table 5, there are no algorithms presented in columns 4-6 or even in the discussion.

Responses: We did not propose different algorithms. The other algorithms used to compare were proposed in other papers. These algorithms were not described directly in our manuscript last time. According to your comments, these algorithms are all described directly (in both Table 6 and the discussion) in the revised manuscript. And the sentence “Finally, we use a parameter-optimized MPGA-ER to solve for more FJSP instances and demonstrate its effectiveness by comparing it with other algorithms.” is changed to “Finally, we use a parameter-optimized MPGA-ER to solve for more FJSP instances and demonstrate its effectiveness by comparing it with that of other algorithms proposed in other studies.” for more clarity. (see the words in red color in Abstract and Section 3.5 of the version named “Revised Manuscript with Track Changes”)

2. No consistency in writing for example in line 249 “(i) Machine crossover: two integers less than” and in line 251 “(ii) Operation crossover is POX crossover”. The same also in line 264 ‘(i) Machine mutation:” and no number in line 268 “Operation mutation:”

Responses: According to your comments, these descriptions have been revised for consistency. (see the words in red color in Sections 2.3.3 and 2.3.4 of the version named “Revised Manuscript with Track Changes”)

3. No consistency in writing for example in line 313 “Finally, the measures: average shortest path (ASP) and size of the maximum connected sub-graph (SMCS) are used to measure the ER network structures.”. However, the authors started describing SMCS first.

Responses: According to your comments, these descriptions have been revised for consistency. (see the words in red color in Section 2.4 of the version named “Revised Manuscript with Track Changes”)

4. In section 2.3, point (3) mutation: different crossover operators were presented and the caption of Figure 2 is Fig 2. Schematic diagram of a crossover. Which one of them is that?

Responses: The crossover operator you mentioned is the operation crossover. And the caption is changed to “Fig. 4. Schematic diagram of operation crossover.” in the revised manuscript. (see Fig. 4)

5. In abstract:”we study how the sub-population size and number and the propagation rate of advantageous genes affect the performance (evaluated by the average optimal value and success rate based on TIN) of MPGA-ER. Is it appropriate to insert the sentence “(evaluated by the average …) in this way?

Responses: According to your comments, this sentence has been changed to “we study how the sub-population number and size and the propagation rate of advantageous genes affect the performance of MPGA-ER, wherein the performance is evaluated by the average optimal value and success rate based on TIN.”(see the words in red color in Abstract of the version named “Revised Manuscript with Track Changes”)

6. In line 144, there is no reference for the mathematical model.

Responses: According to your comments, we have added two references to the mathematical model. (see the words in red color in Section 2.1 of the version named “Revised Manuscript with Track Changes”)

7. Symbols of FJSP should be tabulated.

Responses: According to your comments, the symbols used in the FJSP model are summarized in Table 2. (see Table 2)

8. The authors did not show a Gantt chart of an arbitrary schedule to the example presented in Table 2.

Responses: According to your comments, we have given a Gantt chart of the schedule represented by F ([2 1 4 4 0 4; 1 1 4 5 4 9; 2 2 5 5 4 9; 1 2 1 1 9 10; 2 3 4 2 9 11]) which is described in Section 2.3.2 in the example in Table 3, shown in Fig. 2. (see Fig. 2)

9. What is Jt in Line 214?

Responses: Jt is the total number of operations of all jobs. According to your comments, we have explained Jt in Table 2. (see Table 2)

10. What is POX crossover stands for?

Responses: POX crossover stands for the operation crossover. For consistency, the word “POX” is deleted in the revised manuscript. And we called it operation crossover directly. (see the words in red color in Section 2.3.3 of the version named “Revised Manuscript with Track Changes”)

11. The two-point crossover was not explained and a figure is needed to present it.

Responses: According to your comments, we have used a figure (Fig. 3) to explain the two-point crossover. (see Fig.3)

12. There is a reference ([27]) in line 251 (“Operation crossover is POX crossover [27]”). This refers to what.

Responses: This reference means that the POX crossover was described in [27]. As described above, the word “POX” is deleted in the revised manuscript, and we called it operation crossover directly for consistency. Then, we explain how to use it, which is also described in [27].

13. Subsections numbers of 2.3 section.

Responses: According to your comments, we have numbered all the subsections. (see Sections 2.3.1~2.3.4)

14. In line 307 it was mentioned” SR, which is also used to evaluate the performance of MPGA-ER”. However, this is the same as that in line 302.

Responses: According to your comments, we have deleted the redundant description. (see Section 2.4)

15. In line 303: “For a given TIN, every algorithm runs several times (denoted by Ntol) independently”. However, this number was not stated.

Responses: Ntol is set to be 10 in this study. According to your comments, this description has been added in the revised manuscript. (see the words in red color in Section 2.4 of the version named “Revised Manuscript with Track Changes”)

16. A short description of how the instances were generated is preferred.

Responses: According to your comments, we have tried to explain why we choose these instances. (see the words in red color in Sections 3.1 and 3.5 of the version named “Revised Manuscript with Track Changes”)

Finally, thank you very much, and your comments are constructive and helpful.

Thank you and beat regards.

Yours sincerely

Xiaoqiu Shi; Wei Long; Yanyan Li; Dingshan Deng

---

## [Decision Letter · Decision Letter 2]

13 May 2020

Multi-population genetic algorithm with ER network for solving flexible job shop scheduling problems

PONE-D-19-22005R2

Dear Dr. li,

We are pleased to inform you that your manuscript has been judged scientifically suitable for publication and will be formally accepted for publication once it complies with all outstanding technical requirements.

With kind regards,

Ashkan Memari

Academic Editor

PLOS ONE

Additional Editor Comments (optional):

Reviewers' comments:

Reviewer's Responses to Questions

**Comments to the Author**

1. If the authors have adequately addressed your comments raised in a previous round of review and you feel that this manuscript is now acceptable for publication, you may indicate that here to bypass the “Comments to the Author” section, enter your conflict of interest statement in the “Confidential to Editor” section, and submit your "Accept" recommendation.

Reviewer #1: All comments have been addressed

Reviewer #2: All comments have been addressed

2. Is the manuscript technically sound, and do the data support the conclusions?

Reviewer #1: Yes

Reviewer #2: Yes

3. Has the statistical analysis been performed appropriately and rigorously? 

Reviewer #1: N/A

Reviewer #2: Yes

4. Have the authors made all data underlying the findings in their manuscript fully available?

Reviewer #1: Yes

Reviewer #2: Yes

5. Is the manuscript presented in an intelligible fashion and written in standard English?

Reviewer #1: Yes

Reviewer #2: Yes

6. Review Comments to the Author

Reviewer #1: Good work. The authors could efficiently address the remaining issues. It can be now accepted for publication.

Reviewer #2: The authors address all the previous comments, and the research work is publishable. I recommend this work for acceptance

7. PLOS authors have the option to publish the peer review history of their article (what does this mean?). If published, this will include your full peer review and any attached files.

Reviewer #1: No

Reviewer #2: Yes: Moath Alatefi

---

## [Editor Report · Acceptance letter]

18 May 2020

PONE-D-19-22005R2 

Multi-population genetic algorithm with ER network for solving flexible job shop scheduling problems 

Dear Dr. Li:

I am pleased to inform you that your manuscript has been deemed suitable for publication in PLOS ONE. Congratulations! Your manuscript is now with our production department. 

With kind regards,

on behalf of

Dr. Ashkan Memari 

Academic Editor

PLOS ONE